# Identification of Genetic Associations of *IDH2*, *LDHA*, and *LDHB* Genes with Milk Yield and Compositions in Dairy Cows

**DOI:** 10.3390/life14101228

**Published:** 2024-09-25

**Authors:** Yu Song, Zhe Wang, Lingna Xu, Bo Han, Dongxiao Sun

**Affiliations:** Key Laboratory of Animal Genetics, Breeding and Reproduction of Ministry of Agriculture and Rural Affairs, National Engineering Laboratory of Animal Breeding, State Key Laboratory of Animal Biotech Breeding, Department of Animal Genetics and Breeding, College of Animal Science and Technology, Agricultural University, Beijing 100193, China; songy@cau.edu.cn (Y.S.); wz123@cau.edu.cn (Z.W.); xulingna@caas.cn (L.X.); bohan@cau.edu.cn (B.H.)

**Keywords:** *IDH2*, *LDHA*, *LDHB*, milk production traits, association analysis

## Abstract

Previous study revealed that isocitrate dehydrogenase (NADP (+)) 2, mitochondrial (*IDH2*), lactate dehydrogenase A (*LDHA*), and lactate dehydrogenase B (*LDHB*) genes were significantly differentially expressed in liver tissues of Holstein cows among different lactation periods and associated with lipid and protein metabolism; hence, they were considered as candidates for milk production traits. Herein, the genetic effects of the three genes on milk yield, fat, and protein traits were studied by association analysis using 926 Chinese Holstein cows from 45 sire families. As a result, five single nucleotide polymorphisms (SNPs) in *IDH2*, one in *LDHA*, and three in *LDHB* were identified by re-sequencing, and subsequently, they were genotyped in 926 Chinese Holstein cows by genotyping by target sequencing (GBTS). With the animal model, single-locus association analysis revealed that four SNPs in *IDH2* and one SNP in *LDHA* were significantly associated with milk, fat, and protein yields (*p* ≤ 0.0491), and three SNPs in *LDHB* were associated with milk yield, milk fat yield, and fat percentage (*p* ≤ 0.0285). Further, four IDH2 SNPs were found to form a haplotype block significantly associated with milk yield, fat yield, protein yield, and protein percentage (*p* ≤ 0.0249). In addition, functional predictions indicated that one SNP in *LDHA*, g.26304153G>A, may affect transcription factor binding and two SNPs, g.88544541A>G and g.88556310T>C could alter *LDHB* mRNA secondary structure. In summary, this study profiled the significant genetic effects of *IDH2*, *LDHA*, and *LDHB* on milk yield and composition traits and provided referable genetic markers for genomic selection programs in dairy cattle.

## 1. Introduction

Milk, as a crucial dietary component in the human diet, serves as a significant source of essential nutrients, providing ample high-quality proteins and energy for the human body [1]. In modern times, there is a growing demand for higher-quality milk due to increased interest in nutrition and better health. This has made improving both the quality and quantity of milk production a pressing concern. The milk production traits in dairy cattle breeding are primarily governed by polygenic regulation involving minor-effect genes [2]. So far, substantial-associated variants have been detected in dairy cattle, with 7411, 18,171, and 20,873 loci for milk yield, protein, and lipid composition, respectively (28 April 2024, http://www.animalgenome.org/cgi-bin/QTLdb/). Only three genes with large to moderate effects, namely diacylglycerol O-acyltransferase 1 (DGAT1) [3,4], growth hormone receptor (GHR) [5], and ATP-binding cassette subfamily G member 2 (ABCG2) [6], were confirmed as predominantly causative genes underlying milk composition in dairy cattle. In the early stage, several candidate functional genes and loci associated with milk production traits were explored by previous studies [7,8,9,10], and further exploration in this field is needed.

Since 2009, the application of genomic selection (GS) has played a pivotal role in addressing the challenge of slow genetic improvement in dairy cattle through the more accurate and earlier selection of individuals with superior milk production characteristics in breeding programs [11]. Previous studies have shown that adding known functional gene information to SNP marker data can improve the accuracy of genomic breeding value prediction [12,13]. Therefore, it is of great significance to mine and screen functional genes affecting milk production traits.

In previous research, transcriptomes and proteomes studies were conducted on liver tissues from Holstein cows during the dry period, early lactation, and peak of lactation and identified nine candidate functional proteins/genes associated with milk production traits, with a particular focus on three genes, isocitrate dehydrogenase (NADP (+)) 2, mitochondrial (*IDH2*), lactate dehydrogenase A (*LDHA*), and lactate dehydrogenase B (*LDHB*) that played critical roles in the glycolytic process, pyruvate and energy metabolism, and the glucagon signaling pathway [14]. IDH2 is involved in the decarboxylation of isocitrate to α-ketoglutarate, a process that is closely linked to fatty acid biosynthesis [15]. Both *LDHA* and *LDHB* genes belong to the lactate dehydrogenase family and are involved in the anaerobic glycolysis process under anaerobic conditions thereby associated with lipid production [16,17,18,19]. In addition, the *IDH2* gene is located near the peaks of the reported quantitative trait loci (QTLs) for milk yield and protein percentage [20,21] with a distance of 0.52~3.69 cM on BAT21. *LDHA* is within the known QTL regions for milk yield (6.93 Mb to the peak) as well as close to the two SNPs, ARS-BFGL-NGS-24998 (0.17 Mb) associated with milk protein percentage and UA-IFASA-8605 (4.35 Mb) associated with milk fat yield, fat percentage and protein percentage identified by GWAS [18]. *LDHB* is 0.31~4.35 Mb to eight SNPs, BTA-10187-rs29015749 (0.31 Mb), BTB-01267305 (0.44 Mb), Hapmap59202-rs29011704 (1.12 Mb), Hapmap60862-rs29018508 (2.34 Mb), BTA-74498-no-rs (2.98 Mb), Hapmap48069-BTA-74468 (3.35 Mb), BFGL-NGS-116999 (4.24 Mb), and BTA-74479-no-rs (4.35 Mb) that were significantly associated with milk traits [22].

Until now, no correlation between these three genes and milk traits has been reported. Consequently, the purpose of this study was to systematically identify genetic variances within the *IDH2*, *LDHA*, and *LDHB* genes and evaluate their impact on milk yield and composition traits in the Chinese Holstein population. Potential functional mutations were proposed, as well as providing valuable genetic markers for genome selection programs.

## 2. Materials and Methods

### 2.1. Animals and Phenotypes Data Collection

The animals used in this study comprised 926 Chinese Holstein cows who were the daughters of 45 sires and were from 22 dairy farms belonging to Beijing Sunlon Livestock Development Co., Ltd. (Beijing, China). All cows were under uniform feeding conditions and conducted regular standardized performance testing for dairy herd improvement (DHI). The body condition score (BSC) is regularly monitored to assess reproductive parameters, including pregnancy rates and calving intervals, to assess fertility. Phenotypic values of 305-day milk yield, fat yield, fat percentage, protein yield, and protein percentage during first (926 cows) and second (632 cows) lactations were provided by the Beijing Dairy Cattle Centre (Beijing, China) and the descriptive statistics of these data are shown in Appendix A.

### 2.2. DNA Extraction and Quality Control

The genomic DNAs from the 45 semen and 926 blood samples were extracted using the salt-out procedures and TIANamp Blood DNA Kits (Tiangen, Beijing, China), respectively. The quantity and quality of the extracted DNA samples were, respectively, measured by a NanoDrop 2000 Spectrophotometer (Thermo Scientific, Hudson, DE, USA) and 1% agarose gel electrophoresis.

### 2.3. SNP Identification and Genotyping

A total of 28 primers (Appendix A) were designed to amplify the entire coding region and 2000 bp of up/downstream flanking regions of *IDH2*, *LDHA* and *LDHB* genes with Primer 3 version 0.4.0 (http://bioinfo.ut.ee/primer3-0.4.0/, accessed on 15 January 2024) based on the genomic sequence of the bovine *IDH2* (GenBank accession no.: NC_037348.1), *LDHA* (GenBank accession no.: NC_037356.1) and *LDHB* (GenBank accession no.: NC_037332.1). The primers were synthesized by Beijing Genomics Institute (BGI, Beijing, China). Two DNA pools were randomly constructed and used for all the polymerase chain reactions (PCR), and each pool had 22–23 semen DNAs with equal concentration (50 ng/μL) per sample. The final reaction volume of PCR included 2 μL genomic DNA (50 ng/μL), 1.25 μL of each primer (10 pmol/μL), 12.5 μL Premix TaqTM (Takara, Dalian, China) and 8 μL RNase-free deionized water (Tiangen, Beijing, China). PCR conditions were as follows: initial denaturation at 94 °C for 5 min, followed by 35 cycles at 94 °C for 30 s, 60 °C for 30 s, 72 °C for 30 s, and a final extension at 72 °C for 7 min. After the amplification, the purified PCR products were bi-directionally sequenced in Beijing Qinke Xinye Biotechnology Co., Ltd. (Beijing, China), and the sequences were analyzed by CHROMAS (version 2.23) and NCBI-BLAST+2.15.0 (https://blast.ncbi.nlm.nih.gov/Blast.cgi, accessed on 20 January 2024) to detect the potential SNPs. The genotyping by target sequencing (GBTS) technology was used to genotype the identified SNPs in 926 cows by Shijiazhuang Breeding Biotechnology Co., Ltd. (Shijiazhuang, Hebei, China).

### 2.4. Estimation of Linkage Disequilibrium

As for the identified SNPs of each gene, the extent of Linkage Disequilibrium (LD) was estimated by the Haploview 4.2 (Broad Institute of MIT and Harvard, Cambridge, MA, USA). The D’ value is proportional to the degree of LD, and haplotypes with frequencies greater than 0.05 were retained.

### 2.5. Association Analysis

The association analyses between SNPs and/or haplotype blocks and the five milk production traits on first or second lactation were conducted by SAS 9.4 mixed procedure using the following animal model:y=µ+HYS+b×M+G+a+e

For each trait, y is the phenotypic value of each cow; μ is the overall mean; HYS is the fixed effect of farm (1–22: 22 farms), year (1–4: 2012–2015), and season (1, April–May; 2, June–August; 3, September–November; and 4, December–March); M is the age of calving as a covariant; b is the regression coefficient of covariant M; G is the genotype or haplotype combination effect; a is the individual random additive genetic effect, distributed as $N\;\left(0,{\;A\delta }_{a}^{2}\right)$ with the additive genetic variance $\delta_{a}^{2}$; and e is the random residual, distributed as $N\;\left(0,{\;I\delta }_{e}^{2}\right)$ with identity matrix I and residual error variance $\delta_{e}^{2}$. Bonferroni correction was applied for multiple testing, and the significant level of the multiple tests was equal to the raw *p* value divided by the number of tests. A statistically significant association was considered distinct from a null effect if the raw *p* value is less than 0.05/n, where n is the number of genotypes or haplotype combinations. Meanwhile, the additive effect (a), dominant effect (d), and substitution effect (α) were calculated as follows: a=(AA−BB)/2; d=AB−(AA+BB)/2; α=a+d(q−p), where, AA, BB, and AB are the least square means of the milk production traits in the corresponding genotypes, p and q are the frequency of allele A and allele B, respectively.

### 2.6. Biological Function Prediction

The JASPAR database (http://jaspar.genereg.net/, accessed on 10 March 2024) was used to predict alterations in transcription factor binding sites (TFBSs) caused by SNPs in the 5′ regulatory regions of the *IDH2*, *LDHA*, and *LDHB* genes (relative score ≥ 0.90).

RNAfold Web Server (http://rna.tbi.univie.ac.at/cgi-bin/RNAWebSuite/RNAfold.cgi, accessed on 25 March 2024) was utilized to predict the changes in mRNA secondary structure for SNPs in untranslated region (UTR) and exon regions. The minimum free energy (MFE) of the optimal secondary structure reflects the stability of mRNA structure. A lower MFE value indicates greater stability in the mRNA structure.

## 3. Results

### 3.1. SNPs Identification in IDH2, LDHA and LDHB Genes

Five SNPs were identified in the *IDH2* gene, one SNP in the *LDHA* gene, and three SNPs in the *LDHB* gene. Specifically, within the *IDH2* gene, two SNPs (g.21496168A>G and g.21494708C>G) were located in the 5′ flanking region, two SNPs (g.21482140C>T and g.21479397C>T) were identified in introns, and one SNP (g.21478496G>A) was present in the 3′ flanking region. For the *LDHA* gene, g.26304153G>A was identified in the 5′ flanking region. In the *LDHB* gene, three SNPs (g.88543276A>G, g.88544541A>G, and g.88556310T>C) were detected in the 5′ flanking region, exon 1, and exon 4, respectively, with g.88544541A>G and g.88556310T>C being synonymous mutations. The genotypic and allelic frequencies of all identified SNPs are summarized in Table 1.

### 3.2. Association Analysis between SNP/Haplotype Block and Five Milk Traits

The genetic association between the nine SNPs of *IDH2*, *LDHA*, and *LDHB* and five milk production traits in dairy cows was analyzed (Table 2). In *IDH2*, two SNPs (g.21496168A>G and g.21494708C>G) were significantly associated with milk, fat, and protein yields in both the first and second lactations (*p* ≤ 0.0491). Two other SNPs (g.21482140C>T and g.21479397C>T) exhibited significant associations on milk, fat, and protein yields in the first lactation (*p* ≤ 0.0102), and with five milk traits in the second lactation (*p* ≤ 0.0058). SNP g.21478496G>A was significantly associated with milk, fat, and protein yields, as well as fat percentage, in the second lactation (*p* ≤ 0.0052). SNP g.26304153G>A in the *LDHA* gene was significantly associated with fat yield in the first lactation (*p* ≤ 0.0146) and milk, fat, and protein yields in the second lactation (*p* ≤ 0.0159). For the *LDHB* gene, SNPs g.88543276A>G and g.88556310T>C displayed significant associations with fat yield in the first lactation (*p* ≤ 0.0285) and with milk yield and fat percentage in the second lactation (*p* ≤ 0.0254). SNP g.88544541A>G was significantly associated with milk yield, fat yield, fat percentage, and protein yield in the second lactation (*p* ≤ 0.0023). Further results on the additive, dominant, and substitution effects of the SNPs in the *IDH2*, *LDHA*, and *LDHB* genes are presented in Appendix A.

The five SNPs in the *IDH2* gene had a strong linkage, forming a haplotype block (D′ = 0.99; Figure 1). The frequency of the four haplotypes, H1 (ACCGA), H2 (ACCCG), H3 (GTTCG), and H4 (GCCCG), were 50.2%, 30.1%, 14.8%, and 4.4%, respectively. Haplotype-based association analysis showed that the haplotype block was significantly associated with milk, fat, and protein yields, and protein percentage in first lactation (*p* ≤ 0.007), and milk and fat yields, and protein percentage in second lactation (*p* ≤ 0.0249; Table 3).

### 3.3. Effects of SNP Mutations on Gene Transcriptional Activity

Changes in transcription factor binding sites (TFBS) were predicted for four SNPs located in the 5′ flanking regions of the *IDH2*, *LDHA*, and *LDHB* genes (Table 4). The allele A of g.21496168A>G of *IDH2* was predicted to create binding site (BS) for transcription factor (TF) ETS1 (relative score (RS) = 0.97), allele G of g.21494708C>G of *IDH2* for TFAP2E (RS = 0.92), allele G of g.26304153G>A of *LDHA* for THAP1 (RS = 0.90), allele A of g.88543276A>G of *LDHB* for PDX1 (RS = 0.90) and HOXA5 (RS = 0.93), and allele G of g.88543276A>G of *LDHB* for GATA1 (RS = 0.95), GATA2 (RS = 0.91) and TCF7 (RS = 0.95).

### 3.4. mRNA Structural Variations Caused by Synonymous Mutation

The secondary structure of mRNA was predicted for two SNPs in the UTR and exon regions of the *LDHB* gene (Table 5). The results indicated that substituting G for A in g.88544541A>G led to a decrease in the MFE of mRNA secondary structure, resulting in increased stability of *LDHB*. Similarly, when T replaced C in g.88556310T>C, the MFE of mRNA secondary structure decreased, leading to enhanced stability of LDHB expression.

## 4. Discussion

Based on previous transcriptomes and proteomes studies in the liver from different lactation periods that identified the *IDH2*, *LDHA*, *and LDHB* genes as promising candidates for milk production traits in dairy cattle, this study further confirmed these genes have significant genetic effects on milk yield and compositions.

The IDH2 is a mitochondrial enzyme that assumes a pivotal role in cellular metabolism by catalyzing the oxidative decarboxylation of isocitrate to yield α-ketoglutarate and NADPH within the Krebs cycle [23]. This enzyme is believed to have critical functions in glucose metabolism, fatty acids metabolism, and glutamine metabolism [24,25]. Research has revealed that *IDH2* knockout results in insulin resistance (IR) and suppressed hepatic lipogenesis and inflammation [26,27]. In ruminants, lactate serves as a crucial glucogenic substrate for gluconeogenesis, and the key enzymes involved in this pathway are lactate dehydrogenase (LDHA and LDHB), which catalyzes the bidirectional conversion of pyruvate and lactic acid [28,29]. In the context of early lactation cows with ketosis, Xu et al. have proposed that the upregulated expression of *LDHA* may prevent excessive loss of adipose tissue, thereby preserving energy reserves during this period [30]. An extensive analysis of *LDHB* expression across various tissues unveiled that the gene is predominantly expressed in adipose tissues, suggesting a probable role for *LDHB* in fat deposition processes [31]. These studies collectively point to the *IDH2*, *LDHA*, and *LDHB* genes as crucial regulators of lipid metabolism, aligning with the results of this study, which demonstrate their significant impact on milk fat traits.

Transcription factors (TFs), a crucial class of protein molecules, may potentially cause variations in gene expression among individuals with different genotypes [32]. In this study, for instance, the TF THP1 disappeared when the allele G mutated to A of g.26304153G>A of the *LDHA*. Previous studies have shown that THAP1 could enhance the transcriptional activity of target genes [33], and the loss of this regulatory effect could explain the phenotypic data, where cows with genotype AA had lower fat yield than those with genotype GG. This suggests that the positive genetic effects of allele G on milk production traits may be due to the activation of *LDHA* expression by THAP1. This finding highlights g.26304153G>A as a potentially critical mutation affecting milk fat traits, warranting further in-depth exploration.

The mRNA secondary structure can regulate gene expression by affecting the stability of RNA molecules, the efficiency of translation, and the activity of regulatory proteins [34,35,36,37]. It was observed that when the allele was either G at g.88544541A>G or T at g.88556310T>C, the mRNA secondary structure stability of *LDHB* was both lower than that of allele A or C, suggesting that the mRNA expression of *LDHB* may be more stable with alleles G or T. Combined with previous studies indicating the possible involvement of the *LDHB* gene in lipid metabolism, these findings suggest that the G or T alleles may be favorable for the development of milk fat traits. In this study, the milk fat phenotype of cows with genotypes GG and TT was relatively higher than that of AA and CC. These results indicate that the enhanced mRNA stability associated with alleles G and T may positively influence *LDHB* expression, ultimately contributing to improved milk fat production. The discovery of these allele-specific effects on gene expression underscores their potential importance in breeding programs focused on enhancing milk fat traits, meriting further research.

## 5. Conclusions

In conclusion, through phenotype–genotype association analysis, this study first demonstrated that the *IDH2*, *LDHA*, and *LDHB* genes have significant impacts on milk yield and composition traits in the Holstein cattle population. The SNP g.26304153G>A in 5′ flanking region regulates the transcriptional activity of the *LDHA* gene by changing the binding site of transcription factor THAP1, and SNPs g.88544541A>G and g.88556310T>C alter the stability of *LDHB* mRNA secondary structure, implying these SNPs may be potential causal mutations. The findings provided valuable genetic markers for genomic selection programs in dairy cattle.

## Figures and Tables

**Figure 1 life-14-01228-f001:**
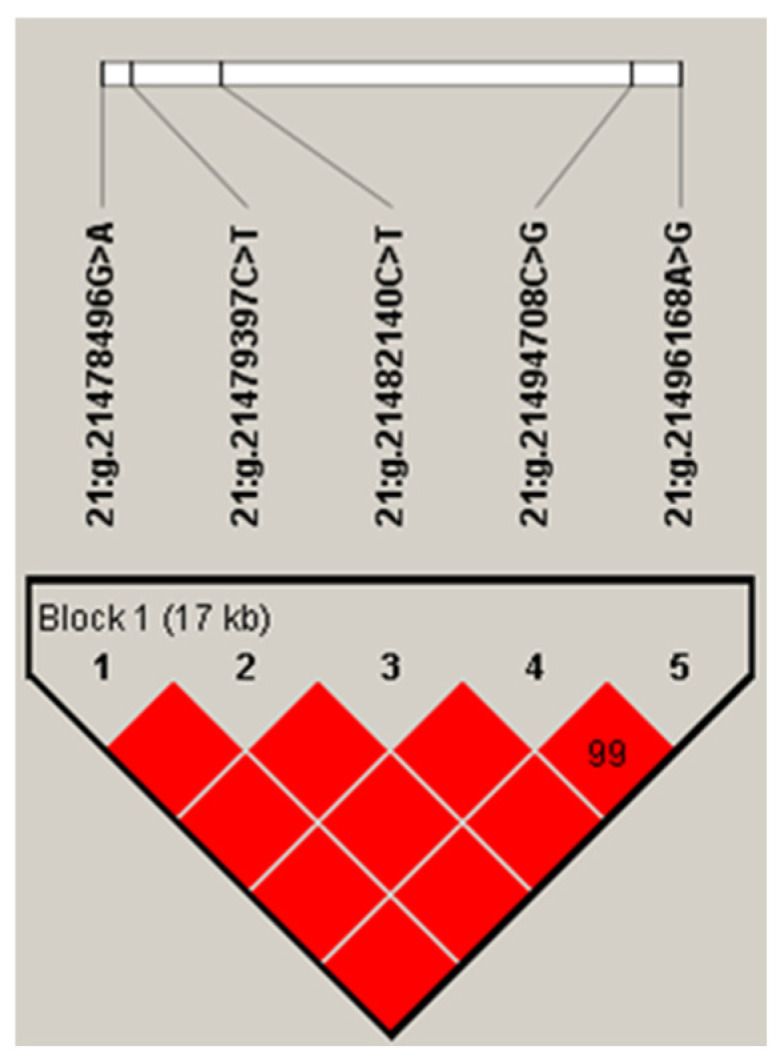
Linkage disequilibrium estimated between SNPs in *IDH2* gene (D′ ≥ 0.99–1.00). The block indicates haplotype block, and the text above the horizontal numbers is the SNP names. The values in boxes are pairwise SNP correlations (D′), while bright red boxes indicate complete LD (D′ = 1).

**Table 1 life-14-01228-t001:** Detailed information of nine SNPs identified in *IDH2*, *LDHA*, and *LDHB* genes.

Gene	SNP	Location	GenBank No.	Genotype	Frequency	Allele	Frequency
*IDH2*	21:g.21496168A>G	5′ regulatory region	rs41970479	AA	0.247	A	0.5043
				AG	0.514	G	0.4957
				GG	0.239		
	21:g.21494708C>G	5′ regulatory region	rs41970478	CC	0.243	C	0.4978
				CG	0.510	G	0.5022
				GG	0.247		
	21:g.21482140C>T	intron	rs444290385	CC	0.717	C	0.8515
				CT	0.269	T	0.1485
				TT	0.014		
	21:g.21479397C>T	intron	rs451726476	CC	0.712	C	0.8488
				CT	0.274	T	0.1512
				TT	0.014		
	21:g.21478496G>A	3′ flanking region	rs41970469	AA	0.636	A	0.8051
				GA	0.338	G	0.1949
				GG	0.026		
*LDHA*	29:g.26304153G>A	5′ flanking region	rs208066940	AA	0.006	A	0.1247
				GA	0.237	G	0.8753
				GG	0.757		
*LDHB*	5:g.88543276A>G	5′ flanking region	rs42924994	AA	0.572	A	0.7570
				AG	0.369	G	0.2430
				GG	0.058		
	5:g.88544541A>G	Exon 1	rs42924993	AA	0.082	A	0.2873
				AG	0.410	G	0.7127
				GG	0.508		
	5:g.88556310T>C	Exon 4	rs41256870	CC	0.296	C	0.5508
				TC	0.510	T	0.4492
				TT	0.194		

**Table 2 life-14-01228-t002:** Associations of the SNPs in *IDH2*, *LDHA*, and *LDHB* genes with milk production traits in two lactations in Chinese Holstein (LSM ± SE).

Gene	SNP	Lactation	Genotype (No.)	Milk Yield (kg)	Fat Yield (kg)	Fat Percentage (%)	Protein Yield (kg)	Protein Percentage (%)
*IDH2*	g.21496168A>G	1	AA (229)	10,084 ^bB^ ± 72.32	338.13 ^bB^ ± 3.14	3.376 ± 0.029	300.96 ^bB^ ± 2.28	2.996 ± 0.019
			AG (476)	10348 ^aA^ ± 60.52	345.32 ^aA^ ± 2.69	3.354 ± 0.024	308.18 ^aA^ ± 1.95	2.989 ± 0.017
			GG (221)	10,265 ^aAB^ ± 73.97	344.08 ^abAB^ ± 3.20	3.369 ± 0.030	307.98 ^aA^ ± 2.33	3.014 ± 0.019
			*p*	0.0004	0.0268	0.6624	0.0009	0.2857
		2	AA (150)	10,499 ^aAB^ ± 73.16	377 ^aAB^ ± 3.15	3.614 ± 0.029	311.96 ^aAB^ ± 2.30	2.979 ± 0.019
			AG (338)	10,597 ^aA^ ± 58.94	377.54 ^aA^ ± 2.62	3.582 ± 0.024	314.12 ^aA^ ± 1.91	2.975 ± 0.016
			GG (144)	10,265 ^bB^ ± 75.52	367.78 ^bB^ ± 3.25	3.603 ± 0.030	304.54 ^bB^ ± 2.37	2.976 ± 0.020
			*p*	<0.0001	0.0029	0.4752	<0.0001	0.9714
	g.21494708C>G	1	CC (225)	10,276 ^aAB^ ± 73.48	343.99 ^abAB^ ± 3.19	3.364 ± 0.030	308.14 ^aA^ ± 2.31	3.011 ± 0.019
			CG (472)	10343 ^aA^ ± 60.59	345.37 ^aA^ ± 2.70	3.357 ± 0.025	308.11 ^aA^ ± 1.96	2.989 ± 0.017
			GG (229)	10,083 ^bB^ ± 72.32	338.14 ^bB^ ± 3.14	3.377 ± 0.029	300.94 ^bB^ ± 2.28	2.996 ± 0.019
			*p*	0.0005	0.0261	0.7475	0.0009	0.3706
		2	CC (147)	10,293 ^bB^ ± 75.03	369.94 ± 3.23	3.612 ± 0.030	305.07 ^bB^ ± 2.35	2.974 ± 0.020
			CG (335)	10,589 ^aA^ ± 59.03	376.75 ± 2.62	3.578 ± 0.024	313.98 ^aA^ ± 1.91	2.976 ± 0.016
			GG (150)	10,498 ^abAB^ ± 73.16	376.92 ± 3.15	3.614 ± 0.029	311.95 ^aAB^ ± 2.30	2.979 ± 0.019
			*p*	0.0002	0.0491	0.3068	0.0001	0.9616
	g.21482140C>T	1	CC (664)	10,255 ^aA^ ± 56.95	343.1 ^aA^ ± 2.56	3.363 ± 0.023	305.7 ^aAB^ ± 1.86	2.992 ± 0.016
			CT (249)	10,323 ^aA^ ± 72.02	345.65 ^aA^ ± 3.12	3.37 ± 0.029	308.83 ^aA^ ± 2.27	3.002 ± 0.019
			TT (13)	9456.31 ^bB^ ± 229.11	309.03 ^bB^ ± 9.31	3.299 ± 0.091	290.38 ^bB^ ± 6.79	3.086 ± 0.054
			*p*	0.0007	0.0004	0.7337	0.0102	0.2003
		2	CC (459)	10,549 ^aA^ ± 55.40	378.29 ^A^ ± 2.49	3.605 ^aA^ ± 0.022	311.65 ^aA^ ± 1.81	2.965 ^bB^ ± 0.016
			CT (168)	10,406 ^aA^ ± 72.59	369.71 ^B^ ± 3.14	3.578 ^aAB^ ± 0.029	312.73 ^aA^ ± 2.29	3.014 ^aA^ ± 0.019
			TT (5)	9143.31 ^bB^ ± 319.77	289.73 ^C^ ± 12.96	3.218 ^bB^ ± 0.128	265.96 ^bB^ ± 9.45	2.909 ^abAB^ ± 0.076
			*p*	<0.0001	<0.0001	0.008	<0.0001	0.0058
	g.21479397C>T	1	CC (659)	10,250 ^aA^ ± 57.05	343.16 ^aA^ ± 2.56	3.365 ± 0.023	305.53 ^bAB^ ± 1.87	2.992 ± 0.016
			CT (254)	10,336 ^aA^ ± 71.62	345.42 ^aA^ ± 3.10	3.363 ± 0.029	309.29 ^aA^ ± 2.26	3.003 ± 0.019
			TT (13)	9461.77 ^bB^ ± 229.18	309.01 ^bB^ ± 9.31	3.297 ± 0.091	290.6 ^cB^ ± 6.79	3.086 ± 0.054
			*p*	0.0005	0.0004	0.7537	0.0054	0.1945
		2	CC (455)	10,546 ^aA^ ± 55.49	378.56 ^A^ ± 2.50	3.609 ^aA^ ± 0.023	311.42 ^aA^ ± 1.81	2.964 ^bB^ ± 0.016
			CT (172)	10,418 ^aA^ ± 72.16	369.11 ^B^ ± 3.13	3.569 ^aAB^ ± 0.029	313.33 ^aA^ ± 2.28	3.015 ^aA^ ± 0.019
			TT (5)	9145.69 ^bB^ ± 319.77	289.53 ^C^ ± 12.96	3.215 ^bB^ ± 0.128	266.13 ^bB^ ± 9.46	2.910 ^abAB^ ± 0.076
			*p*	<0.0001	<0.0001	0.0044	<0.0001	0.0036
	g.21478496G>A	1	AA (589)	10,255 ± 57.86	343.36 ± 2.60	3.366 ± 0.023	305.72 ± 1.88	2.992 ^b^ ± 0.016
			GA (313)	10,289 ± 68.61	343.79 ± 3.01	3.361 ± 0.028	307.29 ± 2.18	2.997 ^b^ ± 0.018
			GG (24)	10,005 ± 170.79	331.24 ± 6.98	3.339 ± 0.068	306.71 ± 5.09	3.083 ^a^ ± 0.041
			*p*	0.2392	0.1828	0.9121	0.6774	0.0744
		2	AA (406)	10,568 ^aA^ ± 56.78	379.41 ^A^ ± 2.54	3.611 ^aA^ ± 0.023	312.15 ^aA^ ± 1.85	2.965 ± 0.016
			GA (212)	10,406 ^bA^ ± 67.50	370.43 ^B^ ± 2.95	3.581 ^aA^ ± 0.027	311.19 ^aA^ ± 2.15	2.997 ± 0.018
			GG (14)	9731.92 ^cB^ ± 195.68	313.55 ^C^ ± 7.97	3.275 ^bB^ ± 0.078	293.49 ^bB^ ± 5.81	3.023 ± 0.046
			*p*	<0.0001	<0.0001	<0.0001	0.0052	0.0683
*LDHA*	g.26304153G>A	1	AA (6)	9840.76 ± 321.37	321.7 ^ab^ ± 13.00	3.290 ± 0.128	295.97 ± 9.48	3.014 ± 0.075
			GA (219)	10,222 ± 73.82	338.27 ^b^ ± 3.20	3.338 ± 0.030	304.38 ± 2.32	2.990 ± 0.019
			GG (701)	10,271 ± 56.59	344.76 ^a^ ± 2.56	3.372 ± 0.023	306.85 ± 1.85	2.998 ± 0.016
			*p*	0.3316	0.0146	0.3754	0.2559	0.8492
		2	AA (6)	9606.23 ^bB^ ± 291.77	334.39 ^bB^ ± 11.82	3.531 ± 0.116	290.44 ^b^ ± 8.62	3.044 ± 0.069
			GA (150)	10,404 ^bAB^ ± 75.13	374.13 ^aA^ ± 3.24	3.616 ± 0.030	309.23 ^a^ ± 2.36	2.978 ± 0.020
			GG (476)	10,542 ^aA^ ± 55.17	376.22 ^aA^ ± 2.48	3.589 ± 0.022	312.49 ^a^ ± 1.81	2.975 ± 0.016
			*p*	0.0015	0.0018	0.5437	0.0159	0.5968
*LDHB*	g.88543276A>G	1	AA (530)	10,227 ± 60.03	341.12 ^b^ ± 2.68	3.358 ± 0.024	305.32 ± 1.95	2.996 ± 0.017
			AG (342)	10,283 ± 65.67	344.61 ^ab^ ± 2.88	3.369 ± 0.026	306.9 ± 2.10	2.998 ± 0.018
			GG (54)	10,402 ± 117.80	353.06 ^a^ ± 4.87	3.389 ± 0.047	310.51 ± 3.55	2.994 ± 0.029
			*p*	0.2714	0.0285	0.7599	0.2773	0.9753
		2	AA (350)	10,443 ^b^ ± 59.36	375.81 ± 2.65	3.627 ^aA^ ± 0.024	310.4 ± 1.92	2.982 ± 0.017
			AG (246)	10,526 ^ab^ ± 64.58	376.08 ± 2.83	3.587 ^aA^ ± 0.026	311.76 ± 2.06	2.974 ± 0.017
			GG (36)	10,806 ^a^ ± 124.81	367.41 ± 5.14	3.390 ^bB^ ± 0.050	318.1 ± 3.75	2.95 ± 0.030
			*p*	0.0135	0.2197	<0.0001	0.1167	0.5363
	g.88544541A>G	1	AA (76)	10,193 ± 102.56	341.4 ± 4.28	3.347 ± 0.041	303.37 ± 3.12	2.981 ± 0.025
			AG (380)	10,324 ± 62.66	344.18 ± 2.76	3.351 ± 0.025	308.21 ± 2.01	2.998 ± 0.017
			GG (470)	10,208 ± 62.02	342.41 ± 2.76	3.378 ± 0.025	304.82 ± 2.01	2.998 ± 0.017
			*p*	0.092	0.647	0.480	0.064	0.753
		2	AA (62)	10,771 ^aA^ ± 100	369.6 ^bAB^ ± 4.17	3.419 ^bB^ ± 0.040	317.57 ^aA^ ± 3.04	2.961 ± 0.025
			AG (262)	10,378 ^bB^ ± 62.45	371.95 ^bB^ ± 2.75	3.603 ^aA^ ± 0.025	307.83 ^bB^ ± 2.00	2.979 ± 0.017
			GG (308)	10,541 ^aAB^ ± 61.73	379.99 ^aA^ ± 2.74	3.633 ^aA^ ± 0.025	313.26 ^aA^ ± 1.99	2.979 ± 0.017
			*p*	0.0001	0.0023	<0.0001	0.0004	0.738
	g.88556310T>C	1	CC (274)	10,349 ± 68.59	348.24 ^A^ ± 2.98	3.378 ± 0.028	309.06 ^a^ ± 2.17	2.997 ± 0.018
			TC (472)	10,232 ± 60.05	340.25 ^B^ ± 2.67	3.345 ± 0.024	304.67 ^b^ ± 1.94	2.990 ± 0.017
			TT (180)	10,182 ± 78.17	342.73 ^AB^ ± 3.36	3.393 ± 0.031	305.86 ^ab^ ± 2.45	3.015 ± 0.020
			*p*	0.0748	0.0059	0.1514	0.0536	0.3327
		2	CC (200)	10,605 ^a^ ± 67.55	376.53 ^AB^ ± 2.94	3.571 ^bB^ ± 0.027	312.39 ± 2.14	2.958 ± 0.018
			TC (311)	10,427 ^b^ ± 59.62	371.45 ^bB^ ± 2.64	3.583 ^bB^ ± 0.024	310.25 ± 1.92	2.985 ± 0.016
			TT (121)	10,515 ^ab^ ± 81.25	385.52 ^aA^ ± 3.49	3.683 ^aA^ ± 0.033	313.71 ± 2.54	2.989 ± 0.021
			*p*	0.0254	<0.0001	0.0019	0.2462	0.1767

Note: The number in the table represents the least squares mean ± standard deviation; the number in the bracket represents the number of cows for the corresponding genotype; *p* shows the significance of the genetic effects of SNPs; ^a^, ^b^ within the same column with different superscripts means *p* < 0.05; and ^A^, ^B^ within the same column with different superscripts means *p* < 0.01.

**Table 3 life-14-01228-t003:** Haplotypes analysis of *IDH2* gene (LSM ± SE).

Lactation	Haplotype Combination (No.)	Milk Yield (kg)	Fat Yield (kg)	Fat Percentage (%)	Protein Yield (kg)	Protein Percentage (%)
1	H1H1 (229)	10,173 ^b^ ± 71.953	342.31 ^B^ ± 3.143	3.375 ± 0.029	304.32 ^cB^ ± 2.289	2.994 ^ab^ ± 0.009
	H1H2 (273)	10,408 ^a^ ± 69.009	348 ^A^ ± 3.0352	3.354 ± 0.028	309.12 ^abAB^ ± 2.209	2.975 ^b^ ± 0.009
	H1H3 (151)	10,402 ^ab^ ± 83.247	349.38 ^A^ ± 3.579	3.368 ± 0.033	312.55 ^aAB^ ± 2.607	3.010 ^a^ ± 0.011
	H2H2 (86)	10,491 ^a^ ± 101.48	357.99 ^A^ ± 4.26	3.411 ± 0.041	315.32 ^aA^ ± 3.104	3.014 ^ab^ ± 0.014
	H2H3 (83)	10,376 ^ab^ ± 101.52	348.55 ^A^ ± 4.256	3.382 ± 0.041	310.53 ^abAB^ ± 3.102	3.001 ^ab^ ± 0.014
	*p*	0.0052	0.007	0.7038	0.0018	0.0154
2	H1H1 (150)	10,681 ^a^ ± 73.169	388.11 ^aA^ ± 3.178	3.635 ± 0.029	319.63 ± 2.314	2.985 ^bB^ ± 0.01
	H1H2 (197)	10,815 ^a^ ± 69.404	391.4 ^aA^ ± 3.047	3.611 ± 0.028	321.94 ± 2.218	2.972 ^bB^ ± 0.009
	H1H3 (106)	10,547 ^b^ ± 82.655	375.95 ^bB^ ± 3.532	3.573 ± 0.033	320.22 ± 2.573	3.031 ^aA^ ± 0.011
	H2H2 (59)	10,646 ^a^ ± 103.91	385.27 ^abAB^ ± 4.357	3.616 ± 0.042	316.28 ± 3.175	2.965 ^bB^ ± 0.014
	H2H3 (51)	10,614 ^a^ ± 107.95	390.85 ^aAB^ ± 4.500	3.689 ± 0.043	321.99 ± 3.280	3.030 ^abAB^ ± 0.015
	*p*	0.0249	0.0003	0.1582	0.4247	<0.0001

Note: LSM ± SE: least squares mean ± standard deviation; the number in the bracket represents the number of cows for the corresponding haplotype; *p* shows the significance level for the genetic effects of SNPs; different superscripts corresponding to the haplotypes indicate significant differences between the haplotypes; ^a^, ^b^ within the same column with different superscripts means *p* < 0.05; and ^A^, ^B^ within the same column with different superscripts means *p* < 0.01.

**Table 4 life-14-01228-t004:** Transcription factor binding sites (TFBSs) prediction for *IDH2*, *LDHA*, and *LDHB* genes.

Gene	SNP	Allele	Transcription Factor (Relative Score ≥ 0.90)
*IDH2*	g.21496168A>G	A	ETS1
		G	
	g.21494708C>G	C	
		G	TFAP2E
*LDHA*	g.26304153G>A	C	THAP1
		T	
*LDHB*	g.88543276A>G	A	PDX1, HOXA5
		G	GATA1, GATA2, TCF7

**Table 5 life-14-01228-t005:** The minimum free energy (MFE) values of optimal secondary structure of *LDHB* mRNA.

SNP	Title 2	Title 3
g.88544541A>G	A	−431.02
	G	−431.39
g.88556310T>C	T	−431.02
	C	−430.97

Note: MFE: minimum free energy.

## Data Availability

The datasets generated and/or analyzed during the current study are available in the article and Appendix A.

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
