# Peer review of "Identification of Genetic Associations of IDH2, LDHA, and LDHB Genes with Milk Yield and Compositions in Dairy Cows"

_life, 2024, doi:10.3390/life14101228_

Round 1
Reviewer 1 Report
Comments and Suggestions for Authors
This manuscript, “ Genetic Effects of IDH2, LDHA and LDHB Genes on Milk Production Traits in Chinese Holstein“ (authored by Song et al.) brings the interesting data of the original study regarding on significant genetic effects of IDH2, LDHA, and LDHB on milk yield and composition traits in Chinese Holstein, which can serve as genetic markers for genomic selection program of dairy cattle.
General comments
The title of this paper is properly defined, but I suggest add „...Cows“ at the end of the title (Genetic Effects of IDH2, LDHA, and LDHB Genes on Milk Production Traits in Chinese Holstein Cows).
The introduction section contains enough references with recent citations (about 1/3 in the last 5 years), as well as other parts of the manuscript. Generally, the study is well-designed and performed with adequate methodology, and the results are particularly novel and of practical value. The conclusions are adequate and of relevance for scientific and practical communities. The use of English is good.
Special comments
1. Can you explain why you chose a different number of months as a season (such as summer and autumn of three months, spring of 2, and winter of 4 months)
2. Please add a note below Table 3, because it is not self-explanatory (e.g. data are presented as LSM ± SE; a, b within the same column with different superscripts means P < ?; and A, B within the same column with different superscripts means P < ?, etc.).
line 264 – please don't use the abbreviation GS (correct: genomic selection)
lines 337-341-please use lowercase for the titles of the manuscripts (the exception of the first letters of the sentences)
line 361- correct abbreviation of the journal Biochem Biophys Res Commun instead Biochemical and Biophysical Research Communications
line 364 – correct the abbreviation of the journal Free Radic Res instead Free Radical Research
line 382 – correct abbreviation of the journal Virus Res instead Virus Research
Author Response
Comments 1: The title of this paper is properly defined, but I suggest add „...Cows“ at the end of the title (Genetic Effects of IDH2, LDHA, and LDHB Genes on Milk Production Traits in Chinese Holstein Cows).
Response 1: Thank you so much for your suggestion. Done as suggested (please see page 1, lines 2-3).
Comments 2: Can you explain why you chose a different number of months as a season (such as summer and autumn of three months, spring of 2, and winter of 4 months)
Response 2: Thanks so much for your comment. Because temperature and humidity can impact milk production, i.e. heat stress reduces milk yield in summer, season has been commonly used as one of fix effects for genetic association analysis and genetic evaluation in dairy cattle. In this study, we considered the months with similar temperature and humidity in Beijing region as one level of season effect thereby improving the accuracy of the analysis.
Comments 3: Please add a note below Table 3, because it is not self-explanatory (e.g. data are presented as LSM ± SE; a, b within the same column with different superscripts means P < ?; and A, B within the same column with different superscripts means P < ?, etc.).
Response 3: Thank you so much for your suggestion. Done as suggested (please see page 9, lines 204-209).
Comments 4: line 264- please don't use the abbreviation GS (correct: genomic selection)
Response 4: Thank you so much for your suggestion. Done as suggested (please see page 9, lines 277-278).
Comments 5: lines 337-341- please use lowercase for the titles of the manuscripts (the exception of the first letters of the sentences)
Response 5: Thank you so much for your suggestion. Done as suggested (please see page 12, lines 357-361).
Comments 6: line 361- correct abbreviation of the journal Biochem Biophys Res Commun instead Biochemical and Biophysical Research Communications
Response 6: Thank you so much for your suggestion. Done as suggested (please see page 13, line 381).
Comments 7: line 364- correct the abbreviation of the journal Free Radic Res instead Free Radical Research
Response 7: Thank you so much for your suggestion. Done as suggested (please see page 13, line 383).
Comments 8: line 382- correct abbreviation of the journal Virus Res instead Virus Research
Response 8: Thank you so much for your suggestion. Done as suggested (please see page 13, line 402).

Reviewer 2 Report
Comments and Suggestions for Authors
Abstract needs to be rewritten-no clear objective and no summary
1. Introduction - you should clearly describe the purpose of the research, what relevance it has for practice
2. Material and methods:
line 81-89 - complete the system of feeding cattle, at what time of the year the research was done ( length of the day), describe the system of keeping animals, condition (BSC), reproductive parameters (fertility of cows) , when were weaned calves.
line 91-94 - when blood samples were taken (time of day)
3. Results - make the tables easier (give the results either in kg or %, there is no need to give in both units
4. The discussion needs to be rewritten, written in general, you should describe the importance of these studies, what new contributions they make to science and practice
5. The conclusion is a repetition of the information from the discussion needs to be improved and written what is the significance of the research , application, practical, what new has been obtained
Author Response
Comments 1: Abstract needs to be rewritten-no clear objective and no summary
Response 1: Thank you so much for your suggestion. Done as suggested (please see page 1, lines 11-29).
Comments 2: Introduction - you should clearly describe the purpose of the research, what relevance it has for practice
Response 2: Thank you so much for the valuable suggestions. As suggested, we have added the purpose of the study and its relevance to dairy breeding practices as following:
“Until now, there is not report on the association between these three genes and milk traits. Therefore, the purpose of this study was to systematically identify genetic variances within the IDH2, LDHA and LDHB genes and evaluate their genetic impacts on milk yield and composition traits, so that providing valuable genetic markers for genomic selection programs in dairy cattle.”(please see page 2, lines 55-57 and lines 78-82 )
Comments 3: line 81-89 - complete the system of feeding cattle, at what time of the year the research was done ( length of the day), describe the system of keeping animals, condition (BSC), reproductive parameters (fertility of cows) , when were weaned calves.
Response 3: Thank you so much for the valuable suggestions. As suggested, we have added the following information about the animals tested in this study as following:
“The animals used in this study comprised 926 Chinese Holstein cows who were the daughters of 45 sires and from 22 dairy farms in Beijing Sunlon Livestock Development Co., Ltd. (Beijing, China) which has conducted regular standardized performance testing (dairy herd improvement, DHI) and information record including body condition score (BSC), date of esrtus, artificial semination, pregnant testing and calving for many years. All cows were fed under the uniform feeding conditions and have been weaned in two months old. Phenotypic values of five milk traits, including 305-day milk yield, fat yield, fat percentage, protein yield and protein percentage in first (926 cows) and second (632 cows) lactations were provided by the Beijing Dairy Cattle Centre (Beijing, China), and the descriptive statistics of these data were shown in Supplementary File S1.” (please see page 2, lines 87-91)
Comments 4: line 91-94 - when blood samples were taken (time of day)
Response 4: Thanks so much for your comment. We collected the blood samples of cows in Septemeber, 2020.
Comments 5: Results - make the tables easier (give the results either in kg or %, there is no need to give in both units
Response 5: Thanks so much for your suggestion. We performed association analyses on five milk production traits, and the results section presents three quantitative traits (in kg) and two rate traits (in %), corresponding to the traits analyzed. Therefore, we believe it is important to retain all five traits in the table for clarity. However, we are open to revising the table if the reviewer prefers simplifying it to a single unit type.
Comments 6: The discussion needs to be rewritten, written in general, you should describe the importance of these studies, what new contributions they make to science and practice
Response 6: Thank you so much for your suggestion. Done as suggested (please see page 10, lines 230-233, lines 248-249, lines 255-260 and line 270-274)
Comments 7: The conclusion is a repetition of the information from the discussion needs to be improved and written what is the significance of the research , application, practical, what new has been obtained
Response 7: Thank you so much for the valuable suggestions. As suggested, we have revised the conclusion as following:
“In conclusion, through phenotype-genotype association analhysis, this study first demonstrated that the IDH2, LDHA, and LDHB genes have significant impacts on milk yield and composition traits in Holstein cattle population. The SNP g.26304153G>A in 5′ flanking region regulates the transcriptional activity of LDHA gene by changing the binding site of transcription factor THAP1, and SNPs g.88544541A>G and g.88556310T>C alter the stability of LDHB mRNA secondary structure, implying these SNPs may be potential causal mutations. Our findings provided valuable genetic markers for genomic selection program in dairy cattle. ” (please see page 10, lines 276-283)
